# Examining Predictors and Outcomes of Decent Work among Chinese Female Pre-Service Primary School Teachers

**Ya Wen** [1], **Huaruo Chen** [2,3], **Xindong Wei** [4,*], **Kai Li** [5], **Fei Liu** [6] and **Xia Liu** [1,7]

1. School of Teacher Education, Nanjing Xiaozhuang University, Nanjing 211171, China
2. College of Education Science and Technology, Nanjing University of Posts and Telecommunications, Nanjing 210023, China
3. Center for Research and Reform in Education, Johns Hopkins University, Baltimore, MD 21286, USA
4. School of Teacher Education, Nanjing University of Information Science &Technology, Nanjing 210044, China
5. School of Philosophy, Wuhan University, Wuhan 430072, China
6. School of Education Science, Nanjing Normal University, Nanjing 210046, China
7. Research Center of Teacher's Ethics Education in the New Era, Nanjing Key Research Base of Philosophy and Social Sciences, Nanjing 211171, China
* Correspondence: weixindong@icloud.com; Tel.: +86-156-952-975-65

**Abstract:** The purpose of this study was to examine the future decent work perceptions in the Psychology of Working Theory (PWT) based on a sample of female primary pre-service teachers in higher education. A cross-sectional design was used for this research. 368 pre-service female teachers were recruited from a university in eastern China for the current study. The results of structural equation modeling indicated that the PWT model applied to Chinese pre-service female primary school teachers. In particular, subjective social status was positively related to future decent work perceptions, disadvantaged mindset was negatively related to future decent work perceptions, and future decent work perceptions were positively related to academic satisfaction. Overall, all the proposed direct pathways in this study were significant. This is the first empirical study to apply the PWT framework among Chinese pre-service female teachers in primary school. With this examination, we were able to comprehensively examine the relationship between variables such as subjective social status and disadvantaged mindset, and future decent work perceptions, helping to improve students' academic satisfaction and providing suggestions for future career development for this group.

**Keywords:** Psychology of Working Theory; decent work; subjective social status; disadvantaged mindset; academic satisfaction; female pre-service primary school teachers

## 1. Introduction

In the past, primary school teachers in China are often considered a form of decent work as the occupation allows individuals to have sufficient time off, receive adequate compensation, and have a high social status [1]. After graduating from high school in China, there are a large number of female students who choose to enter college to study primary education so that they can have the option of becoming primary school teachers after their higher education in college [2]. However, in order to reduce the burden of homework and out-of-school training for students in compulsory education, the Chinese government proposed a "Double Reduction" policy in July 2021, aiming to facilitate students' diversified development through after-school services by teachers [3]. The impact of the "Double Reduction" policy has increased the workload of Chinese primary school teachers by requiring them to work longer hours, which has generated a general job burnout and a low sense of professional identity, especially in the group of female teachers [4]. This current situation of female primary school teachers may adversely affect the professional outlook of female pre-service teachers who are at the college level. In addition, along with the impact

of the global epidemic on employment, the competition to work as a primary school teacher has become increasingly fierce. Therefore, the current perceptions of female pre-service primary teachers in China regarding decent work may be an interesting research topic. Are primary school teachers still decent work in the views of this group?

However, there are relatively few studies on the future perceptions of decent work among female pre-service primary school teachers. The Psychology of Working Theory (PWT) provides a pathway to our questions, which attempts to focus on the decent work of individuals in terms of social background and socioeconomic status [5,6]. Most existing papers on the PWT focus on Western cultural contexts and lack model testing around decent work in a collectivist perspective [7]. Therefore, we would like to examine the perceptions of the group of Chinese female pre-service primary school teachers about the future of decent work from this theory.

## 2. Literature Review

### 2.1. Decent Work

Decent work as the central variable of the PWT has gained the attention of a large number of scholars worldwide [8–12]. PWT is a theoretical model that can be empirically tested based on research in occupational psychology, multicultural psychology, and sociology of work [13]. As a highly inclusive career theory, it aims to explain the work experiences of all individuals, especially individuals who are poor or near poor and who face discrimination and marginalization in their lives [7,14]. PWT takes decent work as the crucial concept and places individuals in a broader social context, examining decent work and human development from the context of social class and economic status [7,15]. Within the framework of PWT, decent work is often defined as having five components: a safe work environment, health care, adequate income, free and rest time, and organizational values that are consistent with family and social values [16,17]. In other terms, decent work is high-quality work that has safe working conditions, adequate pay, adequate health insurance, sufficient time off, and is compatible with family values [18].

Previous studies on decent work have focused on predictors and outcomes [19–21]. On the one hand, several empirical studies have identified antecedent variables of decent work including career adaptability, work volition, economic constraint, marginalization, financial strain, psychological ownership, and work climate [22–26]. For example, a survey based on female workers in the United States revealed that career adaptability, work volition, and workplace climate positively predicted decent work [27]. On the other hand, research by other scholars confirmed that the outcome variables of decent work include survival needs, social contribution needs, self-determination needs, job satisfaction, life satisfaction, well-being, and meaningful work [14,17,24,28–30]. For instance, a study based on Chinese adult workers uncovered that decent work has a positive impact on employee well-being [29]. In addition, decent work can help individuals achieve physical and mental health [30]. Given that the participants in our study are Chinese pre-service primary school teachers who are part of the school population that is not yet employed, this study explores the future decent work perceptions of this group.

### 2.2. Subjective Social Status

Subjective social status is defined as one's perceived relative standing in a society which is derived from social status [31]. The concept of social status refers to the power and control one has over resources and denotes one's relative position in the socio-economic and cultural hierarchy [25]. More specifically, social status is the differences in people's status in social life, depending on the material wealth, income, education, and other social resources that individuals possess and their perceived position in society when compared with others [32]. As can be seen, social status is comprised of two major aspects: objective social status with income, education level and occupation as the measurement indicators; and subjective social status that includes individual self-perception [32,33]. Subjective social status is a subjective perception of oneself based on objective social status, which

refers to where individuals regard themselves in society and may manifest some parts that are more difficult to be reflected by objective indicators [25,32]. Given that our study is more concerned with people's perceived work perception and perceived status, the focus of our study is on subjective social status.

According to the PWT framework, social status is considered to be the main predictor of the process of career advancement based on economic and social resources [7]. Previous research has observed that people of lower social status are more inclined to value work as a means of survival, while people of higher social status are more probable to recognize work as a way to demonstrate their abilities and interests [34]. As research deepens, subjective social status has received increasing academic attention and is regarded as a better predictor of individual career development than objective social status [25]. It has been reported that individuals of higher subjective social status have more life satisfaction [35]. In a recent series of surveys based on the PWT framework, it became apparent that subjective social status, which is a key variable in shaping an individual's concept of work, significantly predicts an individual's work volition, work satisfaction, work meaning, and career adaptability [18,25,32,36,37]. For instance, an examination based on Chinese urban workers discovered that subjective social status positively predicted decent work, while decent work positively predicted work satisfaction [25]. In addition, a recent study denoted that the subjective social status of poor Chinese college students indirectly predicted perceptions of future decent work through work volition [18]. Based on the fact that existing studies have not explored the relationship between subjective social class and decent work among pre-service female primary school teachers, this study proposed the first hypothesis:

**Hypothesis 1 (H1):** *Subjective social status positively influences decent work.*

### 2.3. Disadvantaged Mindset

Disadvantaged mindset belongs to a kind of self-graphic, a concept somewhat similar to subjective social status [38]. This is a variable that has recently been raised by scholars that highlight an individual's perception or evaluation of the strengths and weaknesses of the self with a sociocultural influence [39]. Specifically, this concept refers to the psychological cognitive tendency of individuals who usually perceive themselves as the weaker group in society, including the tendency to feel that they have trouble resisting external risks and are at a disadvantage in social competition [38]. Individuals with a disadvantaged mindset tend to view society as full of risks and often perceive themselves as being at a disadvantage in society. It is not easy to cope with the potential risks in life even if one works hard, thus making it difficult to improve the standard of living for oneself and one's family.

As a novel variable concerned with individual subjective mindsets, relatively little research has been conducted on disadvantaged mindsets. For example, some scholars have used college students as subjects and identified that disadvantaged mindset can significantly and negatively predict systemic justice belief [39]. Since its inception, the PWT framework has paid special interest to decent work for vulnerable groups [40,41]. Although the PWT framework has specifically stressed the impact of subjective social status on variables such as decent work [32]. However, no scholar has yet investigated the possible place of the disadvantaged mindset in the PWT framework and how this variable might affect the likelihood of decent work. If scholars investigate the relationship between disadvantaged mindset and decent work, it may help to expand the model of PWT. In addition, a significant disadvantaged mindset for teacher is burnout. Teachers are often high achievers who enjoy working hard and are constantly seeking for ways to improve. These traits often cause teacher burnout. Teacher burnout is a severe kind of persistent stress that can affect any teacher, regardless of experience or enthusiasm for their profession [42]. For this reason, the present research put forward the second hypothesis:

**Hypothesis 2 (H2):** *Disadvantaged mindset negatively influences decent work.*

### 2.4. Academic Satisfaction

Academic satisfaction which can be characterized as satisfaction with the achievement of academic goals or aspirations usually refers to a student's overall satisfaction with his or her academic performance and achievement while in school [43]. College students' academic satisfaction is considered to be an important factor influencing the career preparation of this group [44]. A survey with a sample of college students revealed that academic satisfaction positively predicted career decision-making self-efficacy [45]. It has been revealed that academic satisfaction strongly predicts depression, anxiety, stress, and psychological well-being levels in college students [46]. Positive career-related outcomes such as career adaptability, calling, and work volition have been identified by scholars as predictors of higher academic satisfaction [47–49].

With the development of the PWT framework, some scholars have discussed the relationship between decent work and academically relevant variables, for example, there was evidence that future decent work perceptions of college students can significantly and positively predict academic engagement, while the relationship with academic satisfaction needs further study [50]. Given the positive impact of academic satisfaction on sustained and successful academic behavior, it is particularly crucial to further analyze the impact of future decent work perceptions on academic satisfaction [51]. Therefore, the relationship between decent work and academic satisfaction with different categories of college students as study subjects deserves further exploration. Given the earlier findings, a third hypothesis was formulated in this study:

**Hypothesis 3 (H3):** *Decent work positively influences academic satisfaction.*

Based on the PWT framework and the above hypotheses, we will also test the specific hypothetical model which is shown in Figure 1.

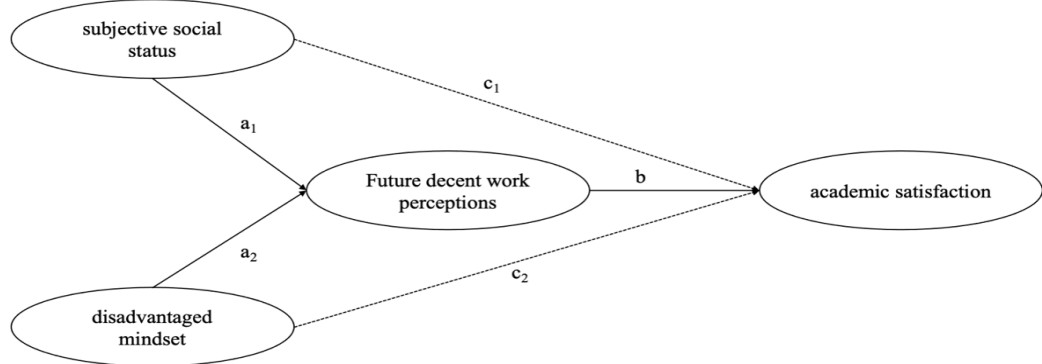

**Figure 1.** The hypothetical model for this study. Note: a1, a2, and b represent direct effect; c1 and c2 are indirect effect.

## 3. Materials and Methods

### 3.1. Participants

All data for this research came from a university in an eastern city in China. We invited the school counselors to assist us with an online questionnaire that we shared with the students. The participants in this study were all pre-service teachers who had a major in elementary education. In this study, 368 valid samples of pre-service teachers were collected. This study's data collection period was from June to August 2021. All participants were informed of the purpose of this study and the confidentiality of their personal information.

### 3.2. Instruments

#### 3.2.1. Subjective Social Status

The subjective social status was evaluated by the MacArthur Subjective Scale of Subjective Social Status [18,31]. This instrument has been demonstrated to reliably predict career outcomes (e.g., career adaptability, work satisfaction, and meaningful work), and applications have been made to samples of Chinese university students [18,25,52]. Participants were presented a drawing of a ladder with 10 rungs, and read "At the top of the ladder are the people who are the best off, those who have the most money, most education, and best jobs. At the bottom are the people who are the worst off, those who have the least money, least education, worst jobs, or no job. Please place an 'X' on the rung that best represents where you think you stand on the ladder". To score this measure, we simply note the number of the rung (1–10) on which the participants placed their "X".

#### 3.2.2. Disadvantaged Mindset

The 8-item Disadvantaged Mindset Scale was used to evaluate disadvantaged mindset [38]. The example items include, "Even if I work hard, it is difficult to cope with the potential risks in life", "Compared with most people, I work harder but I am not more successful", "I have free time during the workweek", and "I have an advantage over most people when it comes to competition [39]". Each participant responded to questions on a 7-point Likert scale which ranged from 1 (Completely Disagree) to 7 (Completely Agree). In the current study, the internal consistency of the total table scores was 0.77. Confirmatory factor analysis results showed that the two-factor model fit the data: $\chi^2 = 46.546$, $p < 0.001$, CFI = 0.952, RMSEA = 0.084, 95% CI [0.059, 0.110], and SRMR = 0.047. Basic information about the scale can be found in Appendix A, Table A1.

#### 3.2.3. Future Decent Work Perceptions

The 15-item Decent Work Scale was designed to assess students' perceptions of future decent work [53]. The scale was comprised of five subscales, namely, workplace safety, health care, adequate compensation, free time and rest, and organizational values aligned with family and social values. We adopted the former research methodology and modified the project to make it more appropriate for our participants [50,54]. Participants were invited to give their answers to these questions based on the instructions below. " Please visualize your future work and respond to the questions based on this image." Each item was reworded to accurately indicate the future direction. For example, "At my future work, I will feel safe from emotional or verbal abuse of any kind (workplace safety); "I will be rewarded adequately for my work" (sound compensation). Students were instructed to score these items from 1 (Strongly Disagree) to 7 (Strongly Agree). The internal consistency coefficient for these items was 0.88. Confirmatory factor analysis results showed that the five-factor model fit the data: $\chi^2 = 304.082$, $p < 0.001$, RMSEA = 0.084, 95%CI [0.074, 0.095], SRMR = 0.065, and CFI = 0.952. Basic information about the scale can be found in Appendix A, Table A2.

#### 3.2.4. Academic Satisfaction

Three items from the 3-item Academic Satisfaction Scale were chosen to assess the academic satisfaction of the participants [50,55]. Participants responded to these items on a 7-point Likert scale ranging from 1 (Strongly Disagree) to 7 (Strongly Agree), with examples including, " I enjoy the level of intellectual stimulation in my course", and " I am generally satisfied with my academic life in my university [51]". The scale showed a strong internal consistency, with $\alpha = 0.79$. This scale has only one factor. Basic information about the scale can be found in Appendix A, Table A3.

### 3.3. Data Analysis

To make sure that the scale has satisfactory psychometric properties, the data were analyzed using SPSS 25.0 to test the bias of the common method and to evaluate the

reliability of the scale by Cronbach's alpha coefficient. To ascertain the suitability of the sample distribution for the ensuing analysis, descriptive statistics were used to quantify the distribution. The correlation among the three variables was tested to evaluate whether the model can be established. A complete model was constructed using the lavaan package in R to examine the relationship among subjective social status, disadvantaged mindset, future decent work perceptions, and academic satisfaction. The lavaan package in R was also used to do the exploratory factor analysis (EFA) and the confirmatory factor analysis (CFA).

## 4. Results

### 4.1. Common Method Deviation Test

Several test scales were used in this study, and all tests were administered uniformly. The content of the scales, the characteristics of the participants, and the testing environment may lead to bias in the results of the study. Therefore, we tested for common method variance using Harman's one-factor test. The first unrotated factor extracted from factor analysis containing all items of interest accounted for 29% (less than 40%) of the total variance [56]. Confirmatory factor analysis results for the scales all showed that the theoretical model fit the data well. The results are shown in the *Instruments* part.

### 4.2. Mean, Standard Deviation, and Correlation Matrix of Variables

The descriptive statistical analysis and correlation matrix for each study variable were shown in Table 1.

**Table 1.** Means, standard deviations, and correlation coefficients of variables (*n* = 368).

| Variable | *M* | *SD* | 1 | 2 | 3 | 4 | 5 |
|---|---|---|---|---|---|---|---|
| 1 Future decent work perceptions | 4.73 | 1.53 | | | | | |
| 2 Subjective social status | 5.32 | 1.37 | 0.31 *** | | | | |
| 3 Disadvantaged mindset | 3.93 | 1.39 | −0.44 *** | −0.32 *** | | | |
| 4 Academic satisfaction | 4.49 | 1.22 | 0.40 *** | 0.10 | −0.27 *** | | |
| 5 Age | 20.48 | 1.27 | −0.10 * | −0.14 ** | 0.04 | 0.14 ** | |
| 6 Urban | 1.50 | 0.50 | −0.11 * | −0.18 *** | 0.09 | −0.07 | 0.05 |

Note: Urban is coded as 1 = urban, 2 = not urban. * $p < 0.05$, ** $p < 0.01$, *** $p < 0.001$.

In the case of this study, we analyzed the data using SPSS 25.0. Correlation analysis demonstrated that subjective social status was positively correlated with future decent work perceptions ($r = 0.31$, $p < 0.0011$; Hypothesis 1), and disadvantaged mindset was negatively correlated with future decent work perceptions ($r = -0.44$, $p < 0.001$; Hypothesis 2). Future decent work perceptions were positively correlated with academic satisfaction ($r = 0.40$, $p < 0.001$; Hypothesis 3). The above findings suggest that participants with low subjective social status also have lower future decent work perceptions. Similarly, the higher the individual's disadvantaged mindset score, the lower the level of future decent work perceptions. Finally, the higher the future decent work perceptions score of the participants, the higher the level of academic satisfaction. Since age was positively correlated with academic satisfaction ($r = 0.14$, $p = 0.006$), we control it when testing the hypothetical model in Figure 1 and the mediation effects.

### 4.3. Final Structural Model and Mediating Effect

For a good model, the criteria for fitting values to the data need to be met, for instance, $\chi^2/df$ values below 3 and RMSEA values below 0.08 [17]. The fit indices for the model in this study were $\chi^2/df$ = 2.547, RMSEA = 0.065, SRMR = 0.078, and TLI = 0.881, CFI = 0.895, all of which are at or close to the critical values. Therefore, we conclude that the model proposed in the previous period has a good fit with the actual data. The final structural equation model was shown in Figure 2.

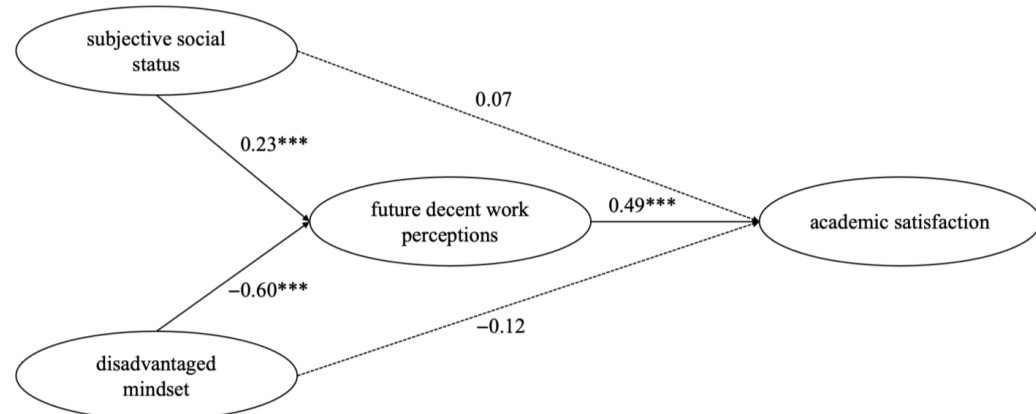

**Figure 2.** Final structural model. Note: *** *p* < 0.001.

A bootstrap 95% confidence interval was produced to inspect the mediation effects. It has been demonstrated in the literature that indirect effects could be confirmed if the confidence interval does not include zero [57]. The results in Table 2 demonstrated that future decent work perceptions significantly mediated the path between subjective social status and academic satisfaction (β = 0.12, *SE* = 0.02, 95%CI [0.013, 0.079]). The effect of future decent work perceptions on academic satisfaction was significantly mediated by disadvantaged mindset (β = −0.29, *SE* = 0.22, 95%CI [−0.927, −0.067]).

**Table 2.** Bootstrap analysis of mediating effect significance test.

| Path | Standardized Indirect Effect | | 95% CI Bootstrap Bias Corrected | |
|---|---|---|---|---|
| | β | SE | Lower | Upper |
| subjective social status→future decent work perceptions→academic satisfaction | 0.12 ** | 0.02 | 0.013 | 0.079 |
| disadvantaged mindset→future decent work perceptions→academic satisfaction | −0.29 ** | 0.22 | −0.927 | −0.067 |

Note: ** *p* < 0.01.

## 5. Discussion

In response to a call to verify the PWT model with a varied sample and to investigate complex propositions in the context of PWT, we collected data from female pre-service teachers in Chinese primary schools. Overall, this study confirmed several interesting findings and added new insights to PWT.

For the predictor component of PWT, our study revealed that, as hypothesized, the subjective social status of pre-service female primary school teachers was positively associated with future decent work perceptions. This implies that the lower the subjective social status that pre-service female primary school teachers perceive themselves to be, the more likely they are to hold negative perceptions about the future of decent work. This result from the present study further supported the findings of previous scholars that lower subjective social status was associated with lower decent work scores, reflecting the cross-cultural heterogeneity of the findings [18,25,58]. In short, the data of the current paper indicated that similar to studies in the United States and Korea, individuals' perceived subjective social status was positively associated with decent work [33,58,59].

H2 indicated that disadvantaged mindset was negatively related to future decent work perceptions, which suggested that those pre-service female primary school teachers who felt they were disadvantaged in society more frequently were more likely to have negative evaluations in obtaining decent work in the future. Given that the relationship between these two variables has not been previously explored by scholars, this finding could increase the new horizon of PWT research.

In addition to exploring the antecedent variables in the PWT model, the current investigation attempted to link career-related variables with academic-related variables. A positive correlation was detected between future decent work perceptions and academic satisfaction, as H3 assumed. This was inconsistent with the previous study by Ma et al. and may be due to the different selection of the research samples for the two studies [50]. This study uncovered that the higher the future decent work perceptions, the stronger the academic satisfaction of female pre-service primary school teachers. Perhaps because primary school teachers in China have always been regarded as relatively decent, and although this group's work has become more stressful in the current context of the "Double Reduction" policy, female pre-service primary school teachers remain positive about their future careers, thus making them more probably to be satisfied with their current studies [2,3,60].

Overall, in the relationship between subjective social status and academic satisfaction, future decent work perceptions are partially mediated. That is to say, the subjective social status of female pre-service primary school teachers could be a key explanatory variable for the level of individual academic satisfaction through its influence on future decent work perceptions. Also, future decent work perceptions partially mediated the relationship between disadvantaged mindset and academic satisfaction. In other words, the disadvantaged mindset status of female pre-service primary school teachers could have an impact on academic satisfaction by influencing individuals' future decent work perceptions. However, it is important to note that these findings need to be interpreted with caution, as they were derived from cross-sectional data and lack longitudinal data over time [27].

## 6. Practical Implications

The results of our study have several practical implications for both practitioners and organizations. In order to help pre-service female primary school teachers better prepare for a successful future career, some practical suggestions need to be brought to the attention of the relevant authorities, as listed below.

Numerous research have revealed that women may face more discrimination in the workplace, which has a hindering effect on their career development [27,59]. Especially in the context of facing limited economic resources, many women were more likely to have difficulty obtaining decent work [61,62]. This study discovered that disadvantaged mindset was negatively related to the future decent work perceptions of female pre-service teachers in primary schools. The findings of this survey suggested that the future practice of education may require all types of educators at the university level to help female pre-service teachers to perceive society more positively, to develop a mindset that they can gradually change their current situation through hard work, and believe that they have the potential to obtain decent work.

At the same time, the government and the education sector have to focus on the significant value of future decent work perceptions for the development of female pre-service teachers in primary schools. On the one hand, the government part of it needs to promulgate educational countermeasures to assist students at this stage to establish correct work perceptions at the pre-service stage. On the other hand, the education section needs to provide various supports for the formation of decent work perceptions of elementary school pre-service teachers in all aspects of talent training.

Finally, it has been proposed that college students' academic satisfaction can be improved through interventions [63]. If career counselors, guidance counselors, and other educators at universities expect to improve the academic satisfaction of female pre-service teachers in primary schools, they need to design programs to change the future decent work perceptions of this group [18,50]. As an example, career counselors might try to incorporate disadvantaged mindset into the design of intervention programs and explore various types of interventions to change the disadvantaged mindset, based on which they

can enhance the perception of decent work for female pre-service primary school teachers and thus strengthen the academic satisfaction of this group at the university level [64].

## 7. Limitations and Future Directions

There are several limitations of this study that need to be addressed in the future. First, the cross-sectional design was adopted for this research, meaning that the data for all scales were collected at one point in time and therefore no causal relationships could be established regarding the variables [24,30,58]. Despite this limitation, the present examination provided support for enriching research on the predictors and outcomes of decent work and laid the groundwork for more rigorous research designs in the future. Second, there also existed the potential for single-method bias for this current study [28]. Given that the data in this report were based on participants' self-reports, in-depth information on the predictors and outcomes of decent work for pre-service female primary school teachers was still lacking [23,65]. Third, the sample was selected to represent only a portion of a certain type of female pre-service primary school teachers, which lacked attention to other types of pre-service teachers, resulting in a loss of diversity and richness in the sample. Fourth, we have explored the impact of disadvantaged mindset and subjective social class on decent work in Chinese culture, and perhaps the size and direction of research hypotheses may differ in other cultures [30].

In general, there are several possible directions for upcoming research on decent work. In the first place, there is a call for further additions of longitudinal studies or more intricate experimental designs on decent work in future studies that can assist us in making inferences about the causality of the findings of PWT [33]. As an example, future researchers might be able to select three-, six-, nine-, or even four-year intervals to conduct multiple surveys of female pre-service primary school teachers, so that the process of change in decent work for this group throughout the college years can be traced [22,65–67]. Furthermore, future research could add qualitative or mixed research, for example, using field surveys or in-depth interviews to provide more in-depth, thorough, and comprehensive explanations for the research on the predictors and outcomes of decent work among pre-service female primary school teachers. In addition, a larger and more diverse group of participants needs to be included as part of future research [25,26,30,68]. This current project was conducted with female primary school pre-service teachers, and future researchers can further analyze decent work not only confined to this group, but also need to be expanded with other types of preservice teachers such as secondary school pre-service teachers and male preservice teachers as research participants to test the theoretical model of decent work further. Ultimately, the PWT model with decent work as the central variable may need further expansion. Future research could include additional cultural context variables to assess the relevance of disadvantaged mindset, subjective social status, and decent work across cultures in other countries or regions. In addition, the PWT model can be further extended by adding antecedent variables similar to disadvantaged mindset and posterior variables such as academic satisfaction [58,69]. Therefore, we expect more exploration of the potential role of decent work in individuals' academic and career development by future researchers.

## 8. Conclusions

In conclusion, the current study was an exploration of the predictors and outcomes of decent work in the PWT model, which enriches research in the area of female teachers' career development, using Chinese female pre-service primary school teachers as the research object. Although the present investigation has some limitations, this study reveals the theoretical significance of decent work for Chinese female pre-service primary school teachers and has important practical implications for educators and career counselors engaged in teacher education.

**Author Contributions:** Y.W. was the PI for the project. X.W. and Y.W. developed the questionnaires. Y.W., H.C., K.L. and F.L. collected the data. X.W. and Y.W. developed the analytical plan and did the statistical analyses. Y.W., H.C., X.W. and X.L. interpreted the outcomes of the statistical analysis and wrote the paper. All authors have read and agreed to the published version of the manuscript.

**Funding:** This research was funded by Jiangsu Province General research project of Philosophy and Social Sciences in universities, grant number "2021SJA0488" and "2022SJYB0182"; China Scholarship Council Projects, grant number "202006860031"; Nanjing Xiaozhuang University Scientific Research "Research on the Development of Vocational Experience Curriculum in General High School"; Nanjing University of Posts and Telecommunications Humanities and Social Sciences Research "Research on Students' Academic Guidance from the Perspective of Career Education".

**Institutional Review Board Statement:** The study was conducted in accordance with the Declaration of Helsinki, and approved by the Ethics Committee of Nanjing XiaoZhuang University (protocol code 2021 No.18 and 2021.4.2 of approval).

**Informed Consent Statement:** Informed consent was obtained from all participants included in the study.

**Data Availability Statement:** The data that support the findings of this study are available from the corresponding author. Restrictions apply to the availability of these data, which were used under license for this study. Data are available from the authors with the permission of Nanjing XiaoZhuang University.

**Acknowledgments:** The authors would like to thank the reviewers for their valuable comments.

**Conflicts of Interest:** The authors declare no conflict of interest.

## Appendix A

**Table A1.** Disadvantaged Mindset Scale.

| No. | Items |
|---|---|
| 1 | Even if I work hard, it is difficult to cope with the potential risks in life. |
| 2 | Compared with most people, I work harder but I am not more successful. |

**Table A2.** Decent Work Scale.

| No. | Items |
|---|---|
| 1 | At my future work, I will feel safe from emotional or verbal abuse of any kind. |
| 2 | I will be rewarded adequately for my work. |

**Table A3.** Academic Satisfaction Scale.

| No. | Items |
|---|---|
| 1 | I enjoy the level of intellectual stimulation in my course. |
| 2 | I am generally satisfied with my academic life in my university. |

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
