# Peer review of "Examining Predictors and Outcomes of Decent Work among Chinese Female Pre-Service Primary School Teachers"

_sustainability, doi:10.3390/su15010730_

Round 1
Reviewer 1 Report
Please notice that all comments are written in order to provide constructive feedback that takes into account the type of work usually published in Sustainability.
First of all, let me indicate that the paper cover an interesting topic and it is well written. One element that clearly stands out after reviewing the manuscript is that the study tries to address an ambitious topic. The main issue in this case is that these variables are analysed taking into account only a specific selection of participant, something that the authors describe in the “7. Limitations and future directions” section. Maybe, the contextual nature of the study should be clearly stated, because, otherwise, it instead provides the reader with an impression that this data and analysis can be generalised.
Despite this last part, if the research design and the instruments that are used are robust enough, and the article is well-structured and written.
Author Response
Dear Reviewer 1,
Thanks for your encouragements! We would like to thank you for the careful reading of our manuscript and for providing us with helpful comments to improve its quality. In the revised version of the manuscript we have tried to address all the comments made by you. All these changes have been highlighted in red. Please, find below a point-by-point response to your comments.
1.We agree with you a hundred percent. The selection of female primary pre-service teachers as a specific sample to test and extend the Psychology of Working Theory (PWT) was a major feature of this study. Future researchers may be able to continue to expand the diversity of the sample, thus enriching the research related to PWT.
2.Thank you so much for your warm reminder about our contextual nature of the study. In the first paragraph of the Introduction to the article, the background about the study is described in detail by us, and the text of these descriptions is highlighted in red.
Once again, thank you for your constructive suggestions. Your suggestions are very helpful to the improvement of the article. Thank you again for your valuable advice. We hope that the revised manuscript is accepted for publication in the Sustainability.
Reviewer 2 Report
Thank you for the opportunity to review the manuscript entitled ‘Examining Predictors and Outcomes of Decent Work among Chinese Female Pre-Service Primary School Teachers in Higher Education’ submitted to Sustainability Journal.
The research reported in the manuscript aimed to investigate the future decent work perceptions in the Psychology of Working Theory (PWT) based on a sample of female primary pre-service teachers in higher education. The manuscript deals with a timely topic that is indeed worthy of inspecting through research. The greatest contribution of the current research is on examining how the PWT framework apply among Chinese pre-service female teachers in primary school.
The manuscript text flows nicely, and the theoretical background as the results are compact and pleasurable to read. Generally, this paper presents the results of a very current and important study. The topic is well argument and documented with previous studies from the scientific literature.
In what follows I have listed my minor notions. I hope that authors find them useful for finalising their manuscript.
Regrettably, I recommend the publication of the manuscript with some minor avenues to the authors to further improve their work.
Some recommendations:
• More extensive linking of the findings to previous research cited in the Literature review would be useful in the discussion
· The authors should provide the results with more detailed and clear results. In particular, I would like to see a structure, a classification of the results that follows the formulation of the research questions. The results could be presented in a more clear and specific manner (divided into more sections) which makes the paper easier to read and understand.
Author Response
Thanks for your encouragements! We would like to thank you for the careful reading of our manuscript and for providing us with helpful comments to improve its quality. In the revised version of the manuscript we have tried to address all the comments made by you. All these changes have been highlighted in red. Please, find below a point-by-point response to your comments.
1.Your suggestion is excellent! Thank you for your valuable suggestion. We are similar to your opinion that some of the ideas in the Discussion section can be linked to those in the Literature Review. Therefore, these elements regarding the Discussion section are marked in red by us, thus reflecting that the existing findings complement and expand on previous studies.
2.Thank you very much for the warm reminder. We added a correspondence between the hypothesis and the results in the results section. In the Results section, all confirmed hypotheses are highlighted in red so that they can be easily viewed by you.
Thank you again for your creative suggestions. Your recommendations are very beneficial for the article improvement. We hope that the revised manuscript is accepted for publication in the Sustainability.
Reviewer 3 Report
Dear Authors,
Please take into consideration the following issues:
1. The TITLE doesn't seem appropriate. It would be better to change the TITLE and introduce the words "In a Higher Education Institution" as "368 pre-service female teachers were recruited from a university in eastern China for the current study".
2. The terms "higher education institution", "disadvanteged mindset", "academic satisfaction"and "subjective social status" should be added to KEYWORDS, and eventually, replace others.
3. The research model and the hypotheses should be moved from the LITERATURE REVIEW to the METHODOLOGY.
4. In the METHODLOGY the authors state that "In this study, 368 valid samples of pre-service teachers were collected." Is this sample representative? The authors should provide data about the total population.
5. In DISCUSSION it would be better to explain the validity of each hypothesis.
6. In CONCLUSIONS, the authors should better outlined why the outcomes of their study are relevant form a scientific point of view. Also, they should emphasize the relationships with other studies.
7. Some English expressions are not appropriate such as "We were curious about the current perceptions..."
8. The list of REFERENCES may be expanded.
Good luck!
Author Response
Dear Reviewer 3,
Thanks for your encouragements! We would like to thank you for the careful reading of our manuscript and for providing us with helpful comments to improve its quality. In the revised version of the manuscript we have tried to address all the comments made by you. All these changes have been highlighted in red. Please, find below a point-by-point response to your comments.
1. Your suggestions for TITLEare great! Thank you for your valuable feedback! We have read the comments carefully and made changes.
Based on your suggestions and the titles of some published papers, the title of the paper was adjusted by us to: Examining Predictors and Outcomes of Decent Work among Chinese Female Pre-Service Primary School Teachers
some published papers:
[21] Kim, M.; Kim, J. Examining Predictors and Outcomes of Decent Work among Korean Workers. Int J Env Res Pub He 2022, 19, 1100.
[64] Ma, Y.; You, J.; Tang, Y. Examining Predictors and Outcomes of Decent Work Perception with Chinese Nursing College Students. Int J Environ Res Public Health 2019, 17, 254.
2. We think your suggestion for KEYWORDS of the article is a fair point! Based on your excellent suggestions, the keywords have been adjusted and marked in red.
3. We believe that your suggestions are very helpful. In similar articles that have been published, the research model and the hypotheses are either placed in the METHODOLOGY or in the LITERATURE REVIEW. We will adjust it if necessary. Thank you very much for your valuable suggestions.
some published papers:
[18] Wei, J.; Chan, S.H.J.; Autin, K. Assessing Perceived Future Decent Work Securement Among Chinese Impoverished College Students. J. Career Assess. 2022, 30, 3-22.
[64] Ma, Y.; You, J.; Tang, Y. Examining Predictors and Outcomes of Decent Work Perception with Chinese Nursing College Students. Int J Environ Res Public Health 2019, 17, 254.
4. Thank you very much for the warm reminder. In some previous studies, this current sample size is in line with the basic requirements for statistics. Thank you again for your precioussuggestions.
5. Thanks for the feedback! In the RESULTS section, the validity of each hypothesis has been stated. Also, in the DISCUSSION section, the findings are analyzed and synthesized with previous literature.
6. Thank you very much for your great suggestions on the CONCLUSIONS. The association of this study with other studies is explained in the CONCLUSIONS. Also, the progress of this study compared with similar studies is discussed in more detail in the DISCUSSION, PRACTICAL IMPLICATIONS, LIMITATIONS and FUTURE DIRECTIONS.
7. Thank you very much for your valuable advice. For this section, we have adjusted the text and highlighted it in red.
8. Thank you for the warm reminder. References have been added by us and the added parts are marked in red.
Once again, thank you for your constructive suggestions. Your suggestions are very helpful to the improvement of the article. Thank you again for your valuable advice. We hope that the revised manuscript is accepted for publication in the Sustainability.
Reviewer 4 Report
Dear Editor,
The article presents an interesting subject matter, with relevant content and in an appropriate structure. Without going into too much depth, it looks like a great piece of work. And I congratulate the authors for it.
However, some minor formatting suggestions, such as the following. Missing brackets on pg 3, when citing references 18, 25, 32, 36 and 37...
Minor suggestions can also be made on the content. E.g.: include the economic situation of teachers in the social status and decent work for teachers in China. Or allude to the well-known "teacher bournout" within the disadvantaged mentality or academic satisfaction.
Apart from the above, there is a serious deficiency in the work, which needs to be remedied. It refers to the methodological section, specifically to the use of scales, and affects all the scales used:
1. The psychometric properties of the MacArthur scale are not detailed. Worse still, nothing is said if it has been adapted to the Chinese context, in which case it would be neither useful nor valid for its application.
2. The calculation of the internal consistency of the designed disadvantaged mindset scale is not sufficient. It is methodologically insufficient to guarantee the validity of the scale. An analysis of the validation of the scale must be undertaken, based on Exploratory Factor Analysis (EFA) and Confirmatory Factor Analysis. This is unavoidable.
3. The same applies to the rest of the scales, where only the analysis of internal consistency is carried out, using cronbach's alpha coefficient.
Experimental validation of the scales should be done, and this is not just another suggestion in the article, but a necessity for their approval.
The results section seems to me to be too brief and doubtful due to the lack of validation of the scales. It would gain in validity of the findings obtained with the scales, and in the findings themselves on the validation of the scales.
Otherwise, the rest of the sections are very appropriate and relevant, with the exception of the fact that the aforementioned deficiency regarding the experimental validation of the scales is not included in the limitations.
Kind regards
Author Response
Thank you very much for your feedback on the article. We have read the comments carefully and made changes. In accordance with the instructions in your letter, we have uploaded the revised manuscript file. Revisions in the text are highlighted in red. Responses to comments are marked in red and are listed below.Please, find below a point-by-point response to your comments.
- Thank you for the warm reminder. The issue regarding the formatting has been corrected. We should have noted this. We have added this in the paper.
2.Your suggestions on content are great! That’s helpful feedback! Teachers are often high achievers who enjoy working hard and are constantly seeking for ways to improve. These traits often cause teacher burnout. Teacher burnout is a severe kind of persistent stress that can affect any teacher, regardless of experience or enthusiasm for their profession [42] . We add this in the paper.
Reference:
- Chang, ML. An Appraisal Perspective of Teacher Burnout: Examining the Emotional Work of Teachers. Educ Psychol Rev 2009, 21, 193-218.
3.We believe that your suggestions on the scale are very helpful. That’s a fair point! The MacArthur Scale of Subjective Social Statues (MacArthur SSS Scale) is widely used both at home and abroad [18,31] . Participants were presented a drawing of a ladder with 10 rungs, and read “At the top of the ladder are the people who are the best off, those who have the most money, most education, and best jobs. At the bottom are the people who are the worst off, those who have the least money, least education, worst jobs, or no job. Please place an ‘X’ on the rung that best represents where you think you stand on the ladder.” To score this measure, we simply note the number of the rung (1-10) on which the participants placed their “X.” We have added some detail about the MacArthur scale in the paper.
References
[18] Wei, J.; Chan, S.H.J.; Autin, K. Assessing Perceived Future Decent Work Securement Among Chinese Impoverished College Students. J. Career Assess. 2022, 30, 3-22.
[31] Adler, N. E., Epel, E. S., Castellazzo, G., & Ickovics, J. R. (2000). Relationship of subjective and objective social status with psychological and physiological functioning: Preliminary data in healthy white women. Health Psychology, 19(6), 586–592.
- We think your suggestion for the third part of the article on Disadvantaged Mindset Scale is a fair point! Based on your advice, we have made some clarifications and highlighted the font in red. Confirmatory factor analysis results showed that the two-factor model fit the data: χ2 = 46.546, p < .001, RMSEA = .084, 95%CI [.059, .110], SRMR = .047, and CFI = .952.
In addition, your recommendations regarding the Decent Work Scale are fair.We have added the information in the paper! Take the Decent work Scale for instance: The results of confirmatory factor analysis showed that the five-factor model fit the data: χ2 = 304.082, p < .001, RMSEA = .084, 95%CI [.074, .095], SRMR = .065, and CFI = .952.
We use the confirmatory factor analysis to test the construct validity. The scales which we used in the paper were all well-developed and widely used. The construct validity for different scales were shown in the instruments part.
Thanks for the feedback! We hope that the revised manuscript is accepted for publication in the Sustainability.
Reviewer 5 Report
The title of the article overall reflects the content of the paper, but could perhaps be shorter.
Presents a correct structure and clear summary.
The theoretical framework is consistent, current and articulated.
The methodology is described clearly and succinctly.
The main results are described and the conclusions are relevant. Presents limitations.
Author Response
Thank you very much for your valuable advice about the title of the article.That’s a fair point. We agree with you a hundred percent. We have read the comments carefully and made changes.We revise the title: Examining Predictors and Outcomes of Decent Work among Chinese Female Pre-Service Primary School Teachers
Thank you for your precious comments and advice. We hope that the revised manuscript is accepted for publication in the Sustainability.
Round 2
Reviewer 4 Report
Dear editor,
The article was already good, before the suggested changes. And it has improved, even more, after them. I accept it.
Author Response
Dear Reviewer 4,
Thank you very much for your encouragements! We would like to thank you for the careful reading of our manuscript and for providing us with constructive comments to improve its quality.
Once again, thank you for your valuable suggestions. Your suggestions are very helpful to the improvement of the article. Good luck with you.